# Reaction Pathways of Gamma-Valerolactone Hydroconversion over Co/SiO₂ Catalyst

Gyula Novodárszki [ID], Ferenc Lónyi *, Magdolna R. Mihályi [ID], Anna Vikár [ID], Róbert Barthos [ID], Blanka Szabó, József Valyon [ID] and Hanna E. Solt [ID]

Institute of Materials and Environmental Chemistry, Research Centre for Natural Sciences, Magyar Tudósok Körútja 2, 1117 Budapest, Hungary; novodarszki.gyula@ttk.hu (G.N.); mihalyi.magdolna@ttk.hu (M.R.M.); vikar.anna@ttk.hu (A.V.); barthos.robert@ttk.hu (R.B.); szabo.blanka@ttk.hu (B.S.); valyon.jozsef@ttk.hu (J.V.); solt.hanna@ttk.hu (H.E.S.)
* Correspondence: lonyi.ferenc@ttk.hu

**Abstract:** The hydroconversion of γ-valerolactone (GVL) over Co/SiO₂ catalyst proceeds in a complex reaction network, resulting in 2-methyltetrahydrofuran (2-MTHF) as the main product, and $C_4$–$C_5$ alcohol and alkane side-products. The catalyst was shown to contain $Co^0$ sites and Lewis acid ($Co^{2+}$ ion)/Lewis base ($O^{2-}$ ion) pair sites, active for hydrogenation/dehydrogenation and dehydration reactions, respectively. The initial reaction step was confirmed to be the hydrogenation of GVL to key intermediate 1,4-pentanediol (1,4-PD). Cyclodehydration of 1,4-PD led to the main product 2-MTHF, whereas its dehydration/hydrogenation gave 1-pentanol and 2-pentanol side-products, with about the same yield. In contrast, 2-pentanol was the favored alcohol product of 2-MTHF hydrogenolysis. 2-Butanol was formed by decarbonylation of 4-hydroxypentanal intermediate. The latter was the product of 1,4-PD dehydrogenation. Alkanes were formed from the alcohol side-products via dehydration/hydrogenation reactions.

**Keywords:** GVL hydroconversion; Co/SiO₂ catalyst; 1,4-pentanediol intermediate; 2-methyltetrahydrofuran; $C_4$–$C_5$ alcohols; reaction network





## 1. Introduction

As a renewable carbon source for the production of biofuels and biochemicals, lignocellulosic biomass has attracted enormous scientific interest [1–7]. Levulinic acid (LA), produced from the polysaccharide components of lignocellulose (cellulose and hemicellulose), is considered as one of the most important platform molecules [4,5,7]. It can be easily converted in consecutive hydrogenation/dehydration reactions to γ-valerolactone (GVL), which has the favorable functionality and reactivity for use in producing biofuels, and as value-added chemicals and intermediates for the chemical industry [3,8–11]. Catalytic hydroconversion of GVL under optimized reaction conditions can lead to selective formation of 2-methyltetrahydrofuran (2-MTHF), having a relatively high energy density, octane number (>90), and miscibility (60 *v/v*%) with gasoline [3,12,13].

It was suggested that bifunctional catalysts of high metal loading and relatively low acidity can steer the reaction towards selective formation of 2-MTHF [3,12,14]. According to the proposed mechanism, the reaction is initiated by hydrogenation of the GVL carbonyl group to obtain cyclic ester 2-hydroxy-5-methyltetrahydrofuran (2-OH-5-MTHF) intermediate. The ester bond of the intermediate becomes hydrogenated, to provide 1,4-pentanediol (1,4-PD), which is then readily cyclodehydrated to 2-MTHF [3,12,15]. The above consecutive reaction is, however, often accompanied by competing and parallel reactions resulting in the formation of side-products. Some side-products may appear in a significant concentration or even as the dominant product in the product mixture, depending on the catalyst and/or the reaction conditions [16–19]. Analyzing and understanding the formation of side

products can provide valuable insights into the reaction mechanism and help to improve the efficiency of the catalytic process.

The hydroconversion of GVL has been studied over a large variety of supported metal catalysts, both in batch and flow-through reaction systems. Bifunctional catalysts containing noble metal (Pt, Pd) components having high hydrogenation–dehydrogenation activity and solid acid support, with strong Brønsted acid centers, usually show high selectivity for pentanoic acid or its ester derivatives in the temperature range of 200–250 °C and under 10–80 bar total pressure [6,12,15]. In contrast, catalysts containing a metal component having moderate hydrogenation–dehydrogenation activity (e.g., Co) supported on non-acidic support material (SiO$_2$, activated carbon) directed the reaction toward 2-MTHF under similar reaction conditions [15,18].

The production of biofuels, such as 2-MTHF, calls for the application of a flow-through catalytic reactor and the use of a heterogeneous, non-noble metal catalyst. In the present study we report the selective hydroconversion of GVL to 2-MTHF using a flow-through microreactor system and silica-supported Co catalyst. At 200 °C and 40% conversion the catalyst showed relatively high 2-MTHF selectivity (>70%) and formation of side-products, such as pentanols, 2-butanol, pentane, and butane. The side-products can be exploited as solvents and fuel additives [18,19]. To elucidate the formation routes of the side-products, we systematically studied the effects of reaction conditions, such as temperature and contact time. The 1,4-PD intermediate and 2-MTHF product was also used as a reactant to gain further insight into the minor pathways.

## 2. Results

### 2.1. XRPD of Co/SiO$_2$ Catalyst

XRPD patterns of the Co/SiO$_2$ catalyst are shown in Figure 1. The diffractograms show a broad line at around 2θ = ~20°, characteristic for amorphous silica (not shown in Figure 1). The catalyst sample calcined at 500 °C contains crystalline Co$_3$O$_4$ phase (diffraction lines at 2θ = 31.3, 36.9, 38.6, 44.8, 55.7, and 59.4°, ICDD 78-1970) (Figure 1a). The average Co$_3$O$_4$ particle diameter, determined using the Scherrer equation, was 35 nm.

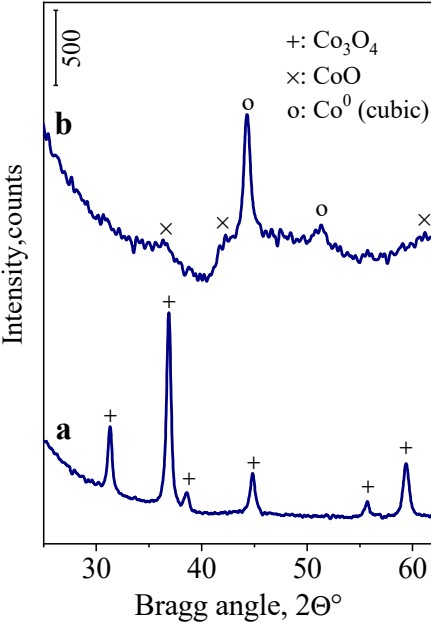

**Figure 1.** XRPD patterns of Co/SiO$_2$ catalyst: (a) air-calcined at 500 °C and (b) reduced in situ at 450 °C in H$_2$ flow for 1 h. Diffractograms were recorded at room temperature.

Reduction of the calcined Co/SiO$_2$ sample in H$_2$ at 450 °C resulted in the formation of Co$^0$ particles having a face-centered cubic (fcc) structure, as indicated by the diffraction

lines at $2\theta = 44.2$ and $51.5°$ (ICDD 15-0806) (Figure 1b). The average diameter of $Co^0$ particles was 27 nm. Note that the TEM image of the reduced $Co/SiO_2$ catalyst (Figure S1) presents homogeneously distributed spherical $Co^0$ particles and agglomerates with a mean diameter of 20–30 nm, which is in good agreement with the average diameter determined by the XRPD method.

Weak broad lines at $2\theta = 36.5, 42.4,$ and $61.6°$ were obtained from the CoO phase (ICDD 48-1719) in the reduced catalyst, indicating that about 10–15% of the $Co_3O_4$ phase was reduced to silica-bound, highly-dispersed CoO (Figure 1b). Note that the XRPD pattern of the used catalyst did not show any significant structural change in the catalyst (Figure S2).

### 2.2. $H_2$-TPR

The $H_2$-TPR curve, measured for the calcined $Co/SiO_2$ catalyst, shows an intense $H_2$-consumption peak in the 300–450 °C temperature range, overlapping with a somewhat broader, low-intensity peak in the range of 450–800 °C (Figure 2a). The total $H_2$ consumption up to 800 °C was 2.60 H/Co, which corresponds to near total reduction of $Co_3O_4$ to $Co^0$ (2.67 H/Co). If the calcined $Co/SiO_2$ sample was pre-reduced in $H_2$ at 450 °C before the TPR measurement (Figure 2b) the $H_2$ consumption was 0.33 H/Co, suggesting that about 12% of the Co remained in non- or partly reduced form in the catalyst after the standard reduction procedure applied before the catalytic runs. The reduction of $Co_3O_4$ to $Co^0$ was presumed to proceed in two steps ($Co_3O_4 + H_2 \rightarrow 3CoO + H_2O$ and $CoO + H_2 \rightarrow Co^0 + H_2O$) [20,21]; however, the observed two reduction peaks obviously cannot be attributed to these two reduction steps, since XRPD measurement showed that most of the $Co_3O_4$ could be reduced to $Co^0$ already at 450 °C (Figure 1b). The XRPD results also showed that 10–15% of the $Co_3O_4$ phase was reduced to CoO, which strongly suggest that the high-temperature reduction peak on the TPR curve can be assigned to the reduction of CoO, strongly interacting with the silica surface. These results are in line with those observed for alumina-supported Co catalysts [22,23]. Note that the cobalt cations on the silica surface are Lewis acid sites (vide infra).

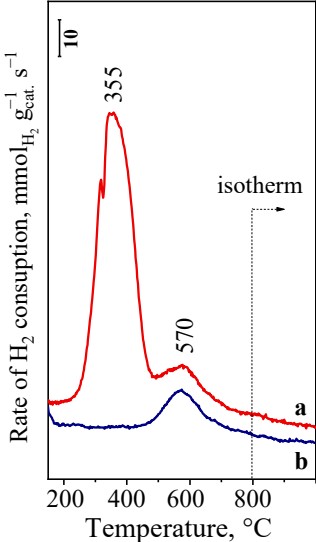

**Figure 2.** $H_2$-TPR curves of the $Co/SiO_2$ catalyst. The sample was either (a) pretreated in $O_2$ flow at 500 °C for 1 h, cooled to 40 °C, and flushed with $N_2$ or (b) pre-reduced in a flow of 9.0 vol% $H_2/N_2$ mixture at 450 °C for 1 h, then cooled to 40 °C in the same gas mixture before the TPR run. The curves were obtained by heating up the sample at a rate of 10 °C·min$^{-1}$ up to 800 °C in a flow of 9.0 vol% $H_2/N_2$ mixture.

### 2.3. CO Chemisorption

The FT-IR spectrum of CO, chemisorbed on the pre-reduced $Co/SiO_2$ catalyst, characterizes the electronic state of the sorption site cobalt species (Figure 3). The $\nu_{CO}$ band at

2006 cm$^{-1}$ is assigned to CO linearly bound to Co$^0$ [24–26]. The relatively broad component band at about 1910 cm$^{-1}$ suggests that bridged carbonyl species, i.e., CO bound to two Co atoms, were also formed [24,26]. The results confirm that the pre-reduced Co/SiO$_2$ catalyst contains Co$^0$ particles that can activate H$_2$ in the GVL hydroconversion reaction.

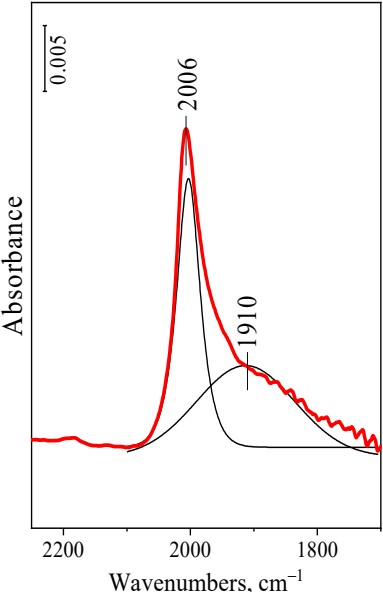

**Figure 3.** FT-IR spectra of CO adsorbed on Co/SiO$_2$ reduced in a H$_2$ flow at 450 °C in situ in the IR cell for 1 h. After reduction, the catalyst was degassed by evacuation, cooled to room temperature, and contacted with CO gas at 5 mbar pressure for 10 min. The CO gas and weakly adsorbed CO was removed by evacuation for 0.5 h, then a sample spectrum was recorded. The thin lines under the curves give the component bands obtained using a peak fitting computer program.

*2.4. FT-IR Spectra of Adsorbed Pyridine*

The FT-IR spectrum of adsorbed pyridine (Py) provides information about the type and concentration of acid sites [27]. The spectra obtained for the pre-reduced Co/SiO$_2$ catalyst and for neat silica support are shown in Figure 4. In line with earlier results [28], only H-bonded Py, giving a pair of bands at 1446/1597 cm$^{-1}$, can be detected on the activated neat silica support (Figure 4(b1)). These bands could be easily eliminated by evacuation at 200 °C (Figure 4(b2)): this is in accordance with the well-documented weak interaction of Py and silanol groups [29,30]. Together with the latter bands, another pair of bands appeared at 1450/1608 cm$^{-1}$ in the spectra of the pre-reduced catalyst (Figure 4(a1–a4)). The bands at 1446/1597 cm$^{-1}$ decreased in intensity when the sample was evacuated at 200 °C, but did not disappear (Figure 4(a1,a2)). This is unlike that found for the catalyst support neat silica. These results suggest that the Py is bound somewhat differently to SiO$_2$ and Co/SiO$_2$, even if it gives bands at the same frequencies (1446/1597 cm$^{-1}$). These and the other pair of bands at 1450/1608 cm$^{-1}$ are assigned to two kinds of Py, coordinately bound to Lewis acid sites [28]. The sorption sites are most probably located at the interface of silica and CoO particles (Figure 1). The bands became gradually weaker as the evacuation temperature was increased, but did not disappear even after evacuation at 400 °C (Figure 4(a2–a4)), indicating the presence of relatively strong Lewis acid sorption sites. The number of sorption sites was estimated from the integrated absorbance of the bands around 1450 cm$^{-1}$ after evacuation at 200 °C (Figure 4(a2)), using the molar extinction coefficient of 1.65 cm µmol$^{-1}$ determined by Zholobenko [31] for coordinately bound Py. The concentration of Lewis acid sorption sites was 10.4 µmol·g$^{-1}$.

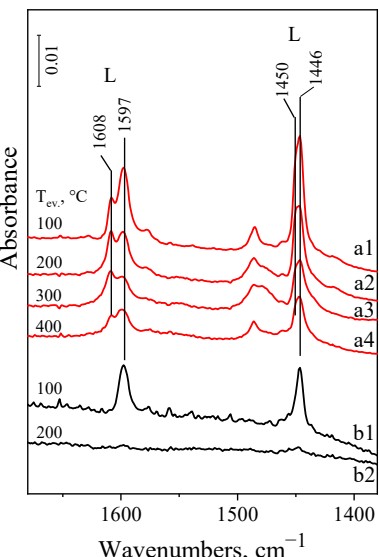

**Figure 4.** FT-IR spectra of pyridine (Py) adsorbed on the (a1–a4) Co/SiO$_2$ catalyst pre-reduced in H$_2$ flow at 450 °C, and on the (b1,b2) pure silica support activated in high vacuum at 450 °C. Pre-treated samples were contacted with 5 mbar of Py vapor at 200 °C for 30 min, cooled to 100 °C and evacuated at the same temperature for 30 min. Evacuation was repeated at temperatures increasing in 100 °C intervals. Spectra were recorded at room temperature. Label L indicates the characteristic bands for Py bonded to Lewis acid sites.

## 2.5. Catalytic Results

### 2.5.1. GVL Hydroconversion

The GVL hydroconversion activity of the Co/SiO$_2$ catalyst was studied by varying the reaction temperature and the spacetime, whereas the total pressure and molar ratio of H$_2$/GVL were fixed at 30 bar and 12, respectively. These conditions were selected because it was found that the pressure dependence of the conversion started to flatten (200 °C) or reached 100% (225 °C) at about 30 bar (Figure S3).

Hydroconversion of GVL was studied over Co/SiO$_2$ at 30 bar total pressure and 1.0 g$_{cat.}$·g$_{GVL}$$^{-1}$·h spacetime in the temperature range of 200–275 °C. Note that the neat SiO$_2$ support showed negligible activity (Table S1). The GVL conversion and product yields are shown as a function of the reaction temperature in Figure 5. The GVL conversion was about 40% at 200 °C (Figure 5A), whereas nearly full conversion was achieved at a somewhat higher reaction temperature (≥225 °C). The main product was 2-MTHF (about 75 mol% selectivity at 200 °C); however, different C$_4$ and C$_5$ alcohol, ketone, and paraffin by-products were also formed, especially at reaction temperature ≥ 225 °C. The maximum yield of 2-MTHF (60 mol%) was reached at 225 °C. Above this temperature, its yield significantly decreased due to acceleration of side-reactions. Among the C$_5$ side-products, the yield of 1-pentanol was somewhat higher than that of 2-pentanol at the lowest reaction temperature (200 °C). At higher temperatures (≥225 °C), 2-pentanol became the dominating side-product (Figure 5A′). Its yield reached its highest value at 250 °C (about 23 mol%) and remained at this level above this temperature. In contrast, the yield of 1-pentanol showed a maximum at 225 °C (about 10 mol%) and then declined to ~2.0 mol% at 275 °C. A minor amount of 2-pentanone (about 2 mol%) also appeared in the product mixture at 225 °C, and its concentration increased linearly up to about 7 mol% as the reaction temperature was increased. The fully deoxygenated side-product pentane appeared at reaction temperature of 225 °C. Its yield increased exponentially as the temperature was increased, reaching about 15 mol% at 275 °C (Figure 5A′). In line with former studies [10,16,32], these results suggest that pentane was formed from pentanols via dehydration and subsequent hydrogenation reactions.

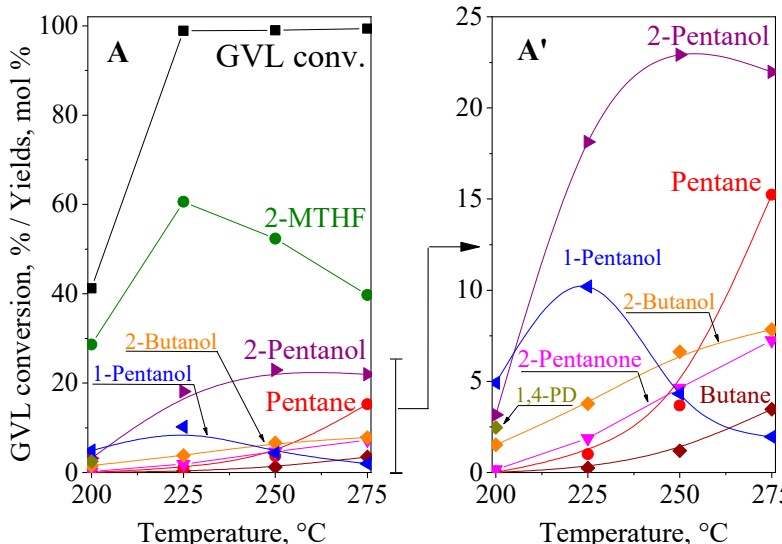

**Figure 5.** GVL conversion and product yields over Co/SiO$_2$ as a function of reaction temperature at 1.0 g$_{cat.}$·g$_{GVL}$$^{-1}$·h spacetime and 30 bar total pressure. The H$_2$/GVL molar ratio was 12. Part (**A′**) shows the lower section of (**A**) enlarged. (2-MTHF: 2-methyltetrahydrofuran).

As a function of temperature, the yield of 2-butanol and butane showed a similar trend to that of 2-pentanol and pentane, but these C$_4$ side-products reached their maximum yields of about 7.0 and 3.5 mol%, respectively, at 275 °C. These yields are significantly lower than those of the C$_5$ products (Figure 5A′).

It should be noted that 1,4-PD, formerly identified as a possible intermediate of 2-MTHF formation [3,12], could be detected only at 200 °C in an amount as small as 2.5 mol% at the applied 1.0 g$_{cat.}$·g$_{GVL}$$^{-1}$·h spacetime (Figure 5A′), whereas it was present in the product mixture in a higher concentration when lower spacetimes were applied (Figure 6A,A′).

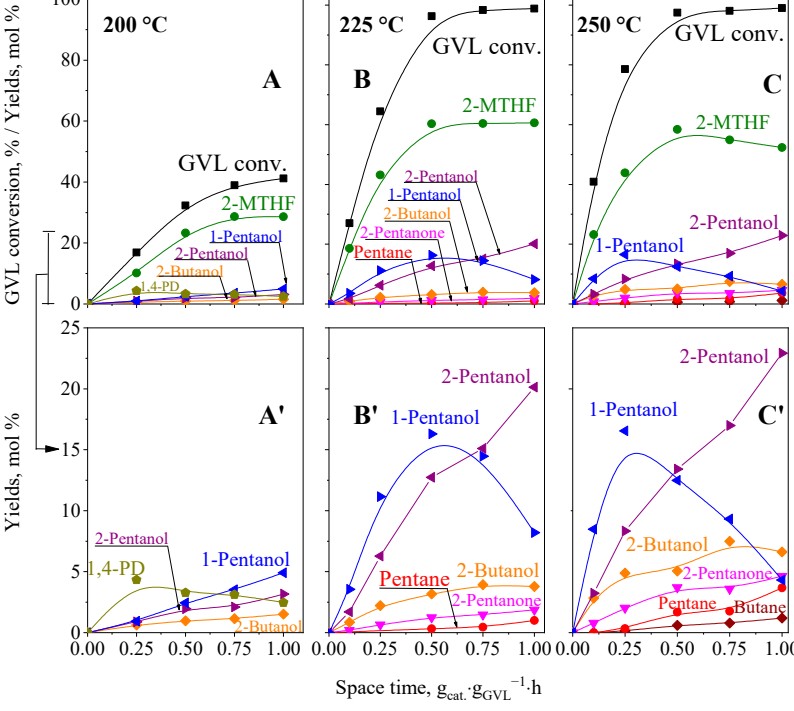

**Figure 6.** GVL conversion and product yields over Co/SiO$_2$ as a function of spacetime at 200 °C (**A**,**A′**), 225 °C (**B**,**B′**) and 250 °C (**C**,**C′**) at a total pressure of 30 bar. The H$_2$/GVL molar ratio was 12. Parts (**A′**–**C′**) show lower sections of (**A**–**C**) enlarged. (2-MTHF: 2-methyltetrahydrofuran).

The conversion and yields increased linearly with the increasing spacetime up to about 40% conversion, except for the yield of 1,4-PD that was passing through a maximum, confirming the notion that it is an intermediate of 2-MTHF formation [3,12]. At the early stage of the reaction, alcohols are presumably formed mainly from the 1,4-PD intermediate rather than from further transformation of 2-MTHF. The possible transformation of 1,4-PD to pentanols via dehydration/hydrogenation reactions was already suggested, e.g., over carbon-supported Ru catalysts in a batch reactor system [10,32].

At higher reaction temperatures ($\geq$225 °C), 1,4-PD intermediate could not be detected in the product mixture, due to its quick transformation (Figure 6B′,C′). At lower spacetimes (0.25–0.5 $g_{cat.} \cdot g_{GVL}^{-1} \cdot h$), where the GVL conversion was not complete, the formation of the 2-MTHF main product and $C_5/C_4$ alcohol side-products followed similar trends to those at 200 °C (cf. Figure 6A–C,A′–C′). However, at higher spacetimes, where the GVL conversion was nearly complete, the yield of 2-MTHF reached about 60% and did not increase further, suggesting that the transformation of the 2-MTHF product also takes place (Figure 6B,C). Interestingly, the yield of the two pentanol isomers changed differently (Figure 6B′,C′). The yield of 1-pentanol exceeded the yield of 2-pentanol in the spacetime range of 0.25–0.5 $g_{cat.} \cdot g_{GVL}^{-1} \cdot h$, then passed through a maximum (~15 mol%) at higher spacetimes, whereas the yield of 2-pentanol increased continuously in the full spacetime range, up to about 23 mol%. These results suggest that at higher temperatures ($\geq$225 °C) and contact times (>0.5 $g_{cat.} \cdot g_{GVL}^{-1} \cdot h$) 2-pentanol could have been formed mostly from 2-MTHF. The appearance of 2-pentanone in the product mixture also supports this notion. The conversion of 2-MTHF was suggested to proceed via cleavage of the [$CH_2$-O] bond, followed by intramolecular hydrogen transfer, to obtain the 2-pentanone intermediate, which was then hydrogenated to 2-pentanol [16,17]. In contrast, the yield of 1-pentanol decreased significantly in the same spacetime range (>0.5 $g_{cat.} \cdot g_{GVL}^{-1} \cdot h$) (Figure 6B′,C′), indicating that the cleavage of the [$CH_3CH$-O] bond of the 2-MTHF ring was less favored. Consequently, the Co/SiO$_2$ catalyst is highly selective in the ring opening of 2-MTHF to 2-pentanol. The hydroconversion study of the 2-MTHF reactant led to a similar conclusion (vide infra).

A small amount of pentane also appeared in the product mixture at higher temperatures ($\geq$225 °C), and its yield increased at increasing spacetimes (Figure 6B′,C′). It was shown that pentane could be formed from pentanols, probably via dehydration and the subsequent hydrogenation reactions [10,16,32].

A minor amount of 2-butanol and butane was also obtained especially at higher reaction temperatures ($\geq$225 °C) and longer spacetimes (Figure 6B′,C′). The alcohol was suggested to form from 1,4-PD in two consecutive reaction steps: dehydrogenation to 4-hydroxypentanal, and decarbonylation. Butane could have been formed by the subsequent dehydration and hydrogenation of 2-butanol [10,32].

### 2.5.2. Hydroconversion of 1,4-PD

In order to further clarify the reaction pathways of GVL hydroconversion, the hydroconversion of the reaction intermediate 1,4-PD was studied (for the first time, according to our best knowledge of the relevant literature) over the Co/SiO$_2$ catalyst (Figure 7). Full conversion could be achieved at long spacetimes, at 200 °C, and already at short spacetimes, at 225 °C (Figure 7A,B). As expected, the main reaction was the cyclodehydration of 1,4-PD to 2-MTHF. The much quicker conversion of 1,4-PD relative to the GVL reactant (cf. Figures 6 and 7) confirms that the rate determining step of the GVL to 2-MTHF reaction over Co/SiO$_2$ is the metal-catalyzed hydrogenation/hydrogenolysis reaction of GVL to produce the 1,4-PD intermediate [32].

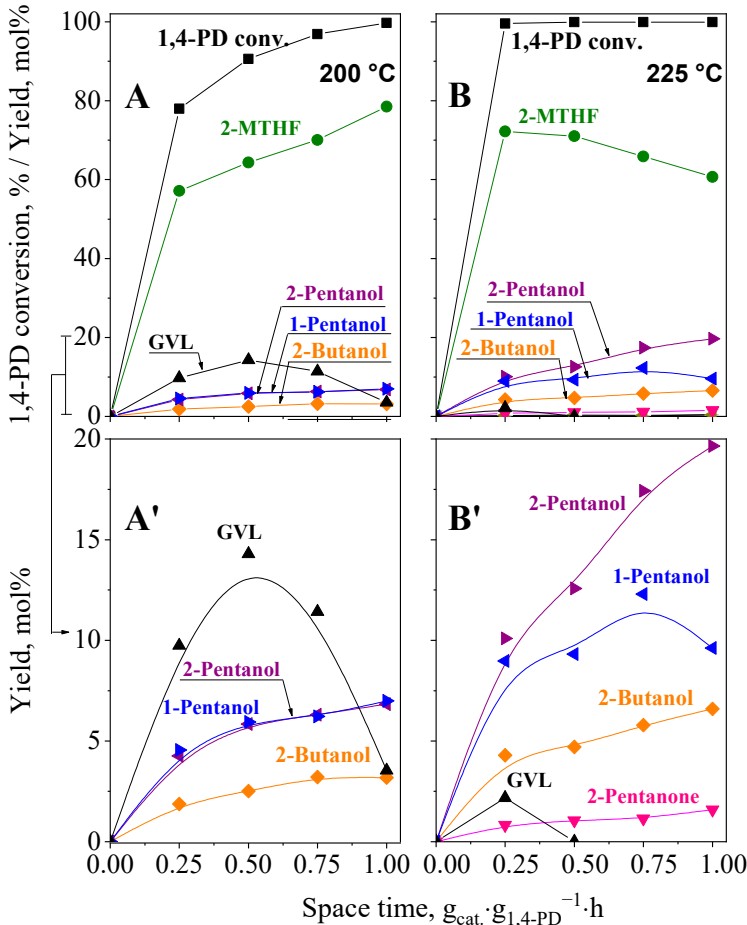

**Figure 7.** 1,4-PD conversion and product yields over Co/SiO$_2$ as a function of spacetime at 200 °C (**A**,**A′**) and 225 °C (**B**,**B′**) at a total pressure of 30 bar. The H$_2$/GVL molar ratio was 12. Sections (**A′**,**B′**) show the lower part of (**A**,**B**) enlarged. (2-MTHF: 2-methyltetrahydrofuran).

The formation of 2-MTHF was accompanied by the formation of 1-pentanol and 2-pentanol side-products (Figure 7A′,B′). At 200 °C, the two alcohols were formed at the same rate (Figure 7A′), and their yields followed the same trend as that of 2-MTHF, suggesting that both pentanols and 2-MTHF were formed from 1,4-PD. At higher temperature (225 °C) the yield of 2-pentanol increased parallel with the spacetime and significantly exceeded the yield of 1-pentanol at longer spacetimes ($\geq$0.25 g$_{cat.}$·g$_{1,4-PD}^{-1}$·h) (Figure 7B′). The 2-pentanol yield increased by about 10 mol% in the spacetime range of 0.25–1.0 g$_{cat.}$·g$_{1,4-PD}^{-1}$·h, which was accompanied by the same decrease in the 2-MTHF yield (cf. Figure 7B,B′). This result suggests that the excess amount of 2-pentanol must have been formed from the 2-MTHF product. Similar conclusions could be drawn from the results of GVL hydroconversion (vide supra).

It was interesting to find that GVL also appeared in the product mixture (Figure 7A′,B′), indicating that the dehydrogenation of 1,4-PD to GVL also proceeded. This reaction is the reverse reaction of the GVL to 1,4-PD reaction, and thus should involve two dehydrogenation steps and presumably proceeds via 2-hydroxy-5-methytetrahydrofuran intermediate. At 200 °C, the yield of GVL had a maximum (13 mol%) at a spacetime of 0.50 g$_{cat.}$·g$_{1,4-PD}^{-1}$·h, and then decreased to 3 mol% (Figure 7A′). However, at 225 °C a negligible amount of GVL was formed. These results suggest that at higher temperatures and spacetimes, the transformation reactions of 1,4-PD, starting with a dehydration step (the formation of 2-MTHF and pentanols), were significantly accelerated relative to the dehydrogenation reaction, leading to GVL formation. As a result, the quick consumption of 1,4-PD decreases the steady-state 1,4-PD concentration and the equilibrium concentration of GVL. It should be noted that at 225 °C and 1.0 g$_{cat.}$·g$_{1,4-PD}^{-1}$·h spacetime, where complete GVL or 1,4-PD

conversions were achieved, about the same 2-MTHF yield (60 mol%) was reached from both reactants (cf. Figures 6B and 7B).

The formation of 2-butanol side-product from 1,4-PD could be also observed (Figure 7A′,B′). This reaction was suggested to proceed via the formation of 4-hydroxypent anal, either by the dehydrogenation of 1,4-PD and/or by the ring opening of 2-hydroxy-5-methytetrahydrofuran intermediate, a transformation which was followed by the decarbonylation of the thus formed 4-hydroxypentanal intermediate to 2-butanol [10,18,32].

### 2.5.3. Hydroconversion of 2-MTHF

Results obtained for the hydroconversion of GVL and 1,4-PD suggested that 2-pentanol could be selectively obtained from the further conversion of the 2-MTHF product. In order to clarify this point, hydroconversion of 2-MTHF was also studied. Results are shown in Figure 8.

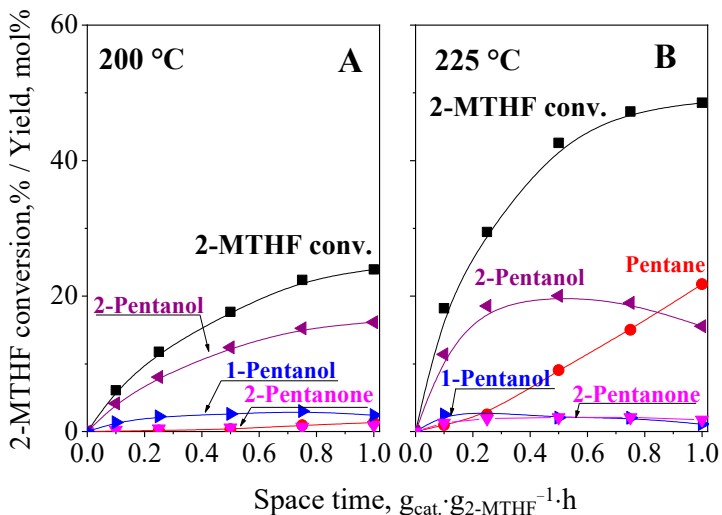

**Figure 8.** 2-MTHF conversion and product yields over Co/SiO$_2$ as a function of spacetime at 200 °C (**A**) and 225 °C (**B**) at a total pressure of 30 bar. The H$_2$/2-MTHF molar ratio was 12. (2-MTHF: 2-methyltetrahydrofuran).

The conversion of 2-MTHF was significantly lower than that of GVL or 1,4-PD reactants under comparable reaction conditions (cf. Figures 6–8). At 200 °C and about 23% conversion pentanols were formed, with high selectivity. Minor amounts of other products, mainly 2-pentanone, were also detected. The yield of 2-pentanol was about 20 times higher than that of 1-pentanol (Figure 8A). These results suggest that the opening of the tetrahydrofuran ring in 2-MTHF at the [CH$_3$CH-O] bond is much less favored than that of the [CH$_2$-O] bond, which is probably due to the hindering effect of the methyl group on the hydrogenolysis of the neighboring bond [17,32]. At a higher reaction temperature (225 °C), the conversion of 2-MTHF reached about 50%, while a significant amount of pentane appeared in the product mixture (Figure 8B). The yield of both pentanols passed through a maximum, indicating that pentane was formed in a consecutive reaction from both pentanols, mainly from the most abundant 2-pentanol (Figure 8B). Note that a significantly higher amount of pentane was formed in the hydroconversion reaction of 2-MTHF than from the reaction of GVL under comparable reaction conditions (cf. Figures 6B′ and 8B). The conversion of GVL involves more reaction steps and surface intermediates that that of the 2-MTHF. It is conceivable that the 2-MTHF coverage of the active sites and the rate of 2-MTHF conversion is higher when instead of GVL, 2-MTHF is the reactant.

### 3. Discussion

In agreement with recent studies [3,12,15], the present work confirms that hydroconversion of GVL over the Co/SiO$_2$ catalyst is initiated by hydrogenation of the GVL

carbonyl group. It was shown that GVL took up one molecule of $H_2$ in the first reaction step, resulting in 2-OH-5-MTHF, which was then further hydrogenated to 1,4-PD key intermediate [15].

The cyclodehydration reaction of 1,4-PD to obtain 2-MTHF is well documented to proceed over both Brønsted and Lewis acid sites [33–36]. The characterization of our $Co/SiO_2$ catalyst revealed that the catalyst contains $Co^0$ sites (metallic Co particles) of hydrogenation/dehydrogenation activity and Co-oxide sites of dehydration activity. The Co-oxide represent Lewis acid ($Co^{2+}$ ion)/Lewis base ($O^{2-}$ ion) pair sites, wherein $Co^{2+}$ ions are in strong interaction with the silica support [37,38]. The cyclodehydration of 1,4-PD can be similarly visualized, as it was carried out over the Lewis acid sites of $\gamma$-$Al_2O_3$ [33,36] (Scheme 1).

**Scheme 1.** Activation and cyclodehydration of 1,4-PD to 2-MTHF over Lewis acid ($Co^{2+}$ ion)/Lewis base ($O^{2-}$ ion) pair sites of $Co/SiO_2$.

In the adsorption complex shown in Scheme 1, the oxygen atom of the primary hydroxyl group is coordinated to the cobalt cation (Lewis acid) and its hydrogen atom to the adjacent oxide anion (Lewis base), whereas the secondary hydroxyl group is coordinated to the surface-bound hydrogen atom. A water molecule is formed by breaking the O–H bond of the primary hydroxyl group and the C–O bond at the secondary hydroxyl group.

The concomitant formation of a new C–O bond results in the formation of a furan ring. The Lewis acid/Lewis base pair sites are regenerated when the products, 2-MTHF and the water, are desorbed. Note that the activation of the secondary hydroxyl group and the elimination of the primary hydroxyl group is also possible in a similar process; however, the cleavage of the secondary hydroxyl group is energetically more favorable than that of the primary hydroxyl group [34].

It was shown that 1,4-PD could be a common intermediate of the 2-MTHF main product and pentanol formation (Figures 6 and 7). Formation of pentanols from 1,4-PD was suggested to proceed via dehydration/hydrogenation reactions [32,36]. The first dehydration step proceeds on acid sites and produces an unsaturated alcohol intermediate, which is then hydrogenated on nearby $Co^0$ sites to pentanol. The activated surface complex formed on Lewis acid ($Co^{2+}$)/Lewis base ($O^{2-}$) pair sites, leading to pentanol formation, should be different from this, resulting in the formation of cyclic ether 2-MTHF. This process is envisioned as shown in Scheme 2.

The adsorption complex is formed via coordination of the hydroxyl oxygen and hydrogen atoms to a $Co^{2+}$ cation (Lewis acid) and oxide anion (Lewis base) surface site, respectively, whereas the neighboring methylene group is coordinated to the surface-bound hydrogen atom (Scheme 2). A water molecule is released via hydrogen transfer from the methylene group (in line with Zaitsev's rule) and a double bond is formed. Removal of the primary hydroxyl group of 1,4-PD leads to 4-penten-2-ol (Scheme 2A), whereas the elimination of the secondary hydroxyl group gives 3-penten-1-ol (Scheme 2B). The unsaturated alcohols are then hydrogenated to pentanols on active $Co^0$ sites. Note that no unsaturated alcohols were detected in the product mixture, possibly due to their facile hydrogenation to saturated alcohols. Transformation of 1,4-PD to pentanols competes with the cyclodehydration reaction, giving 2-MTHF. The former reaction proceeds via a high-energy intermediate unsaturated alcohol in two steps, requiring both acidic and hydrogenation catalytic functions. Therefore, the formation of 2-MTHF prevails (Scheme 3).

Results of the 1,4-PD conversion experiments (Figure 7A) also revealed that a hydrogenation/dehydrogenation equilibrium existed between GVL and 1,4-PD, under the applied reaction conditions (Scheme 3). However, GVL did not appear in the product mixture under reaction conditions which favored the rapid conversion of 1,4-PD to main and side-products (Figure 7B).

1,4-pentanediol                4-penten-2-ol

**A**

1,4-pentanediol                3-penten-1-ol

**B**

**Scheme 2.** Activation and dehydration of 1,4-PD to 4-penten-2-ol (**A**) and 3-penten-1-ol (**B**) over Lewis acid (Co$^{2+}$ ion)/Lewis base (O$^{2-}$ ion) pair sites of Co/SiO$_2$.

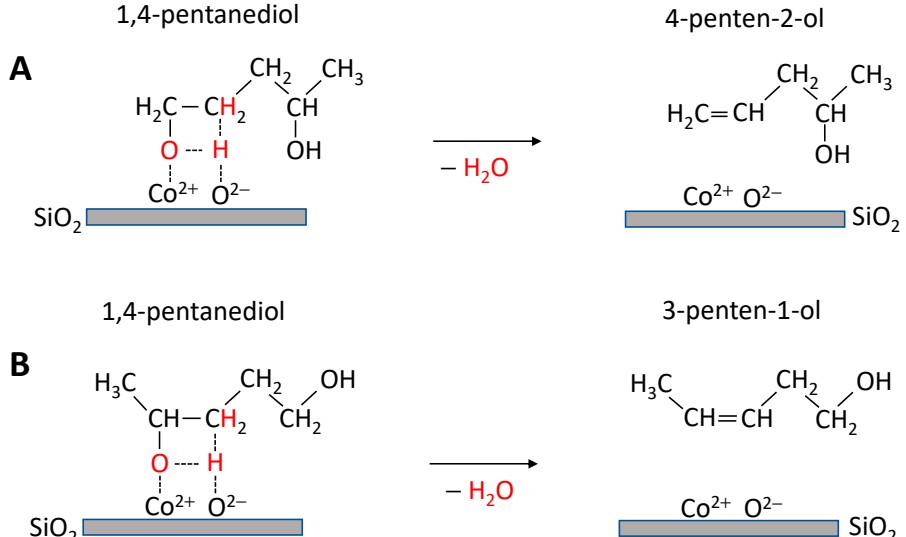

**Scheme 3.** Proposed pathways for the hydroconversion of GVL over Co/SiO$_2$ catalyst, including formation of 2-MTHF main product and C$_4$–C$_5$ alcohol and alkane side products.

Using GVL (Figure 6) or 1,4-PD (Figure 7) as a reactant, the yield of 2-pentanol significantly exceeded the yield of 1-pentanol at reaction temperatures $\geq$ 225 °C and at spacetimes over about 0.5 g$_{cat.}$·g$_{GVL\ or\ 1,4-PD}^{-1}$·h. Results strongly suggested that part of the 2-pentanol exceeding the amount of 1-pentanol came from further transformation of the 2-MTHF. The catalytic results proved, in contrast to that found by Iino et al. [17], that the ring opening of 2-MTHF was more favored at the [CH$_2$-O] bond than at the [CH$_3$CH-O] bond of the furan ring, most probably because the methyl group decreased the reactivity of the neighboring C-O bond. Thus, the ring-opening reaction results favorably in 2-pentanone. 2-Pentanone was a detected intermediate that was hydrogenated to 2-pentanol (Scheme 3). Results also confirmed that a pentane side-product was formed from pentanols in consecutive dehydration/hydrogenation steps.

Our results showed that the side-product 2-butanol could be formed in the hydro-conversion reaction of both GVL and 1,4-PD (Figures 6 and 7). The reaction was suggested to proceed via the 4-hydroxypentanal intermediate. This intermediate can be formed either by dehydrogenation of 1,4-PD and/or by the ring opening of 2-hydroxy-5-methyltetrahydrofuran intermediate [10,18,32]. Note that we could not detect 4-hydroxypentanal as a product, probably due to its rapid decarbonylation to 2-butanol (Scheme 3).

## 4. Experimental

### 4.1. Catalyst Preparation

The silica-supported Co catalyst was prepared by wet impregnation. The support material (CAB-O-SIL EH-5 type, Cabot Corp., specific surface area 385 $m^2 \cdot g^{-1}$) was impregnated with a 0.4 M aqueous solution of $Co(NO_3)_2 \cdot 6H_2O$ (Sigma-Aldrich, >98%, St. Louis, MO, USA) using 4 $cm^3$ solution per gram of silica support and then dried at 110 °C for 12 h, and calcined at 500 °C for 3h. The calcined sample, containing 7.8 wt% Co (determined by ICP-OES analysis), is referred to as $Co/SiO_2$. The specific surface area of the $Co/SiO_2$ catalyst (determined by the BET method) was 303 $m^2 \cdot g^{-1}$.

### 4.2. Catalyst Characterization

The Co content of the catalyst was determined by ICP-OES analysis (Perkin-Elmer, Waltham, MA, USA).

Specific surface area was calculated by the BET method from $N_2$ adsorption isotherms measured at −196 °C, using an automatic gas adsorption instrument (SURFER, Thermo Fisher Scientific, Waltham, MA, USA). The sample was pretreated by evacuation at 350 °C for 12 h before the adsorption isotherm was determined.

The X-ray powder diffractograms (XRPD) of the crystalline phases of the catalyst were recorded by a Philips PW 1810/3710 diffractometer (Philips, Amsterdam, The Netherlands), equipped with graphite monochromator and an X-ray tube operated at 40 kV voltage and 35 mA current. The XRPD patterns were obtained using $CuK_\alpha$ (λ = 0.1541 nm) radiation and by applying a scan step size of 0.02 degrees 2-theta, while the scan time was five seconds in each step. The reduced $Co/SiO_2$ sample was prepared by reducing the calcined sample in situ at 450 °C in $H_2$ flow for 2 h, using a type HT1200 Anton Paar high-temperature sample chamber, before the XRPD measurement. The average crystallite size of the metal oxide or metal particles formed was determined by the Scherrer equation, evaluating the FWHM values of the diffraction lines by applying a full profile fitting method.

The morphology and distribution of the metal particle size on the catalysts were characterized by transmission electron microscopy (TEM) (Philips, Eindhoven, The Netherlands). Diluted aqueous suspensions of samples were drop-dried on carbon-coated copper TEM grids, and the images were taken using a Morgagni 268D microscope (100 kV, W filament, point resolution = 0.5 nm).

The reducibility of cobalt species formed in the calcined $Co/SiO_2$ sample was studied using the method of temperature-programmed reduction, using hydrogen as the reducing gas ($H_2$-TPR) (home-made apparatus). About 100 mg of the calcined sample was placed in a quartz reactor tube (6 mm ID) and was pretreated in a 30 $cm^3 \cdot min^{-1}$ $O_2$-flow at 500 °C for 1h, then cooled to 40 °C and flushed with $N_2$ for 10 min. To obtain the $H_2$-TPR curve, the sample was exposed to a 20 $cm^3 \cdot min^{-1}$ flow of 9.0 vol% $H_2/N_2$ mixture, and then the temperature was ramped up at a rate of 10 °C·$min^{-1}$, up to 800 °C. A second $H_2$-TPR curve was also measured to determine the hard-to-reduce fraction of the Co species in the catalyst. In this case, the calcined sample was pre-reduced under the same conditions applied during the above $H_2$-TPR measurement, except that the heating was stopped at 450 °C for 1h and then the sample was cooled to 40 °C, before starting a new $H_2$-TPR measurement. The reactor effluent was passed through a trap cooled by liquid nitrogen (−196 °C) to remove water from the gas flow. The rate of hydrogen uptake was followed by monitoring the $H_2$ concentration of the reactor effluent, using a thermal conductivity detector (TCD) next to the trap.

The acidity of the $Co/SiO_2$ catalyst sample was reduced at 450 °C, and that of the pure silica support (for comparison) was characterized by the infrared (IR) spectra of pyridine (Py) adsorbed on the Brønsted and/or Lewis acid sites of the catalyst [27]. IR spectra of the sample were recorded by a Nicolet Impact Type 400 Transmission Fourier-transform infrared (FT-IR) spectrometer (Thermo Scientific, Waltham, MA, USA) prior to, and after Py adsorption, using an in-situ cell and applying the self-supported wafer technique. The wafer of the $Co/SiO_2$ catalyst was pretreated in situ in the $O_2$ flow at 500 °C for 1 h, then reduced in $H_2$ at 450 °C for 1h, and then evacuated in high vacuum ($10^{-6}$ mbar) before the experiment, whereas the reduction step was omitted for the pure silica support. The activated sample was contacted with Py vapor at 5 mbar and 200 °C for 30 min, then cooled to 100 °C, degassed at this temperature for 30 min, and cooled to room temperature. The degassing step was repeated at 200, 300, and 400 °C. Spectra were recorded at room temperature, averaging 128 scans at a resolution of 2 cm$^{-1}$. All spectra were normalized to wafer thickness of 5 mg·cm$^{-2}$. Spectral characteristics of adsorbed Py were obtained by subtracting the spectrum of the wafer from the spectrum of the Py-loaded wafer.

The electronic state of the cobalt in the reduced $Co/SiO_2$ catalyst was characterized by the FT-IR spectrum of CO adsorbed on the metal particles [39]. The CO adsorption experiments were carried out in the same way as the Py adsorption experiments, except that the reduced catalyst was cooled down to room temperature before it was contacted with CO gas at 5 mbar pressure for 10 min. The spectrum was taken after the CO gas and weakly bound CO were removed from the cell by evacuation at room temperature to $10^{-6}$ mbar for 30 min. The difference spectrum gave the spectrum of surface species obtained from CO adsorption.

*4.3. Catalytic Measurements*

Catalytic hydroconversion of GVL was studied using a high-pressure flow-through microreactor (ID: 12 mm). The reactor was loaded with 1 g of $Co/SiO_2$ catalyst (0.315–0.630 mm size sieve fraction). The calcined catalyst sample was reduced in situ in the $H_2$-flow of 100 cm$^3$ min$^{-1}$ at 30 bar and 450 °C for 2 h, prior to catalytic runs. The GVL reactant was fed into the reactor using a high-pressure micro metering pump (Gilson, Model 302) (Gilson, Medison, WI, USA), whereas the hydrogen flow was controlled by a mass flow controller (Brooks Instruments, Hatfield, PA, USA). The $H_2$/GVL molar ratio and the total pressure were held constant at 12 and 30 bar, respectively, whereas the reaction temperature was varied in the range of 200–275 °C. The spacetime of GVL was varied between 0.1 and 1.0 $g_{cat}·g_{GVL}^{-1}·h$. Hydroconversion of 1,4-PD and 2-MTHF was also investigated. These experiments were carried out at 200 and 225 °C, 30 bar total pressure and $H_2$/reactant molar ratio of 12, whereas the spacetime was varied between 0.1 and 1.0 $g_{cat.}·g_{reactant}^{-1}·h$. The reactor effluent leaving the reactor via a back pressure regulator was separated into liquid and gaseous products in a water-cooled atmospheric separator. The gaseous products were analyzed using an on-line GC (Varian 3300) equipped with an FID detector and a 30 m long alumina/chloride capillary column (Supelco). The liquid products, collected in each hour, were analyzed using a GC-MS apparatus (Shimadzu QP2010 SE, Shimadzu Corp., Tokyo, Japan) equipped with a 30 m ZB-WAX PLUS capillary column. The experimental error of product analysis and the carbon balance calculated from the feed and that of the reactor effluent was within ±5% in each catalytic run.

## 5. Conclusions

The $Co/SiO_2$ catalyst had to be pre-reduced to become active in the hydroconversion of GVL. Partial reduction of the silica-supported cobalt oxide phase generated $Co^0$ sites, active in hydrogenation/dehydrogenation reactions. The oxide residue represents Lewis acid ($Co^{2+}$ ion)/Lewis base ($O^{2-}$ ion) pair sites, active in dehydration. It was confirmed that 1,4-PD is a key intermediate in the network of transformations. Competing reactions of cyclodehydration and dehydration/hydrogenation led to the 2-MTHF main product, and 1-pentanol and 2-pentanol side-products, respectively. A small amount of 1,4-PD could have

been dehydrogenated to the 4-hydroxypentanal intermediate. This intermediate could not be detected, probably because it readily decarbonylated to the detected product 2-butanol. Under reaction conditions where consecutive ring-opening/hydrogenation transformations of 2-MTHF proceeded, mainly 2-pentanol was formed. Therefore, the concentration of 2-pentanol could significantly exceed the concentration of 1-pentanol. Alkanes were formed via dehydration/hydrogenation reactions from the alcohol side-products.

**Supplementary Materials:** The following supporting information can be downloaded at: https://www.mdpi.com/article/10.3390/catal13071144/s1, Figure S1: TEM image of the reduced $Co/SiO_2$ catalyst, Figure S2: XRPD pattern of the used $Co/SiO_2$ catalyst. Sample was used in the hydroconversion of GVL to 2-MTHF for 58 h time-on-stream. Reaction conditions during the catalytic run were 225 °C, 30 bar, and space time of 1 $g_{cat.} \cdot g_{GVL}^{-1} \cdot h^{-1}$, Figure S3: GVL hydroconversion and product yields over $Co/SiO_2$ catalysts as a function of $H_2$ partial pressure at 200 °C (A) and 225 °C (B) and 1 $g_{cat.} \cdot g_{GVL}^{-1} \cdot h$ space time. The partial pressure of GVL was kept constant at 2.3 bar. 2-MTHF: 2-methyltetrahydrofuran; 1,4-PD: 1,4-pentanediol; Table S1: GVL conversion over $SiO_2$ support at different reaction temperatures. Table S2: Weisz-Prater criterion and Thiele modulus [40,41] for the HDO reaction of GVL on $Co/SiO_2$ at 2.3 bar GVL partial pressure and 225 °C at increasing contact times.

**Author Contributions:** Conceptualization, G.N., F.L. and H.E.S.; investigation, G.N., A.V., H.E.S., R.B. and B.S.; data curation, M.R.M., R.B. and B.S.; writing—original draft preparation, G.N., F.L. and M.R.M.; writing—review and editing, F.L. and J.V.; visualization, A.V., H.E.S. and B.S.; supervision, F.L.; project administration, M.R.M., F.L. and G.N.; funding acquisition, M.R.M., G.N. and J.V. All authors have read and agreed to the published version of the manuscript.

**Funding:** This research was co-financed by the Ministry of Innovation and Technology of Hungary from the National Research, Development and Innovation Fund, under the 2019-2.1.13-TÉT_IN funding scheme (Project No. 2019-2.1.13-TÉT_IN-2020-00043), by the Ministry for Culture and Innovation from the source of the National Research, Development and Innovation Fund, under the ÚNKP-22-4 New National Excellence Program (Eötvös Loránd University) (G.N.).

**Data Availability Statement:** Not applicable.

**Conflicts of Interest:** The authors declare no conflict of interest. The funders had no role in the design of the study; in the collection, analyses, or interpretation of data; in the writing of the manuscript; or in the decision to publish the results.

## Abbreviations

| | |
|---|---|
| 1,4-PD | 1,4-pentanediol |
| 2-MTHF | 2-methyltetrahydrofuran |
| 2-OH-5-MTHF | 2-hydroxy-5-methyltetrahydrofuran |
| FID | Flame ionization detector |
| FT-IR | Fourier-transform infrared spectroscopy |
| FWHM | Full width at half maximum |
| GVL | γ-valerolactone |
| $H_2$-TPR | Temperature-programmed reduction by hydrogen |
| ICP-OES | Inductively coupled plasma optical atomic emission spectroscopy |
| Py | pyridine |
| TCD | Thermal conductivity detector |
| TEM | Transmission electron microscopy |
| XRPD | X-ray powder diffraction |

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
