# Peer review of "Reaction Pathways of Gamma-Valerolactone Hydroconversion over Co/SiO2 Catalyst"

_catalysts, doi:10.3390/catal13071144_

Round 1
Reviewer 1 Report
The paper presents pathways of gamma-Valerolactone hydroconversion on Co/SiO2 catalyst. It is a topic of interest to the researchers in the related areas. This is a well-written paper containing interesting results which merit publication. However, a number of points need clarifying and certain statements require further justification. There are given blow:
1. When explaining the formation pathways of by-products including pentanol, 2-butanol, pentane, and butane, have the factors affecting different solvents been considered?
2. How to prove complete removal of carbon monoxide gas in Figure 3?
3. Is there any other pathway for the activation and dehydration of 1,4-PD to 2-MTHF, 4-penten-2-ol (A), and 3-penten-1-ol (B) over the Lewis acid (Co2+ion)/Lewis base coordination point of Co/SiO2 in scheme 1 and scheme 2?
4. The mass fraction of Co has a significant impact on Co/SiO2 catalysts. When drawing the conclusion that 1,4-PD is a key intermediate in the transformation network, should the performance of the Co/SiO2 catalyst with gradient Co mass fraction used in the article be examined first?
Author Response
Reviewer#1
The paper presents pathways of gamma-Valerolactone hydroconversion on Co/SiO2 catalyst. It is a topic of interest to the researchers in the related areas. This is a well-written paper containing interesting results which merit publication. However, a number of points need clarifying and certain statements require further justification. There are given blow:
- When explaining the formation pathways of by-products including pentanol, 2-butanol, pentane, and butane, have the factors affecting different solvents been considered?
In the present work, the hydroconversion of GVL was studied in a fixed-bed, flow-through reactor using pure GVL without any solvent in the reaction system. The reaction proceeded in gas phase under the applied reaction conditions.
- How to prove complete removal of carbon monoxide gas in Figure 3?
The infrared cell containing the pellet of the catalyst sample was evacuated to 10–6 mbar before introducing 5 mbar of CO into the cell. After 10 min contact time, the gas phase CO (as well as weakly adsorbed CO) was removed by evacuation for 0.5 h. The pressure of the cell reached 10–6 mbar within a few seconds after starting the evacuation, which clearly indicated that no CO gas remained in the cell. Note that CO gas, if present in the cell could be easily detected in the collected spectrum. The experimental section was slightly modified.
- Is there any other pathway for the activation and dehydration of 1,4-PD to 2-MTHF, 4-penten-2-ol (A), and 3-penten-1-ol (B) over the Lewis acid (Co2+ion)/Lewis base coordination point of Co/SiO2in scheme 1 and scheme 2?
Concerning the cyclodehydration of 1,4-PD to 2-MTHF in scheme 1: As we mentioned in the text (see page 10), the activation of the secondary hydroxyl group and the elimination of the primary hydroxyl group is also possible in a similar process; however, the cleavage of the secondary hydroxyl group is energetically more favorable than that of the primary hydroxyl group (see Ref. [34]).
Concerning the dehydration of 1,4-PD to unsaturated alcohols in scheme 2: The leaving group is either the primary (A) or the secondary (B) OH-group; however, according to Zaitsev’s rule, the hydrogen atom must come from the adjacent carbon having the fewest hydrogen substituent, that is from the methylene group. This rule was considered in scheme 2. The text explaining scheme 2 was slightly modified.
- The mass fraction of Co has a significant impact on Co/SiO2catalysts. When drawing the conclusion that 1,4-PD is a key intermediate in the transformation network, should the performance of the Co/SiO2catalyst with gradient Co mass fraction used in the article be examined first?
In a former study (see Ref. [11]) we prepared silica supported Co catalysts containing 4 – 14 wt% Co. Characterization results suggested that the optimum Co content was around 8 wt% concerning the hydrogen uptake and metal dispersion. In the present study, the Co/SiO2 catalyst containing about 8 wt% Co was proved to have suitable hydrogenation activity to direct the reaction toward 2-MTHF main product. It is generally accepted in the relevant scientific literature that formation of 2-MTHF proceeds via 1,4-PD intermediate (see e.g., in Refs. [3] and [12]) and our results are in full agreement with this notion (see Fig. 6 in the manuscript).
Reviewer 2 Report
In this manuscript, the authors carried on reaction evaluation of GVL hydrogenation utilizing pre-reduced Co/SiO2, discovered that partial reduction of the silica-supported cobalt oxide phase generated Co0 sites, active in hydrogenation/dehydrogenation reactions and the oxide residue represents Lewis acid (Co2+ ion)/Lewis base (O2- ion) pair sites, active in dehydration. As far as I'm concerned, there are still some points that could be further enhanced. So, I suggest a revision of the manuscript before publication. The recommendations are listed as below:
1)→The background introduction is a little overly brief, I hope that the background introduction can be more concise and detailed.
2)→In the Abstract part , please cite the detailed data of product distribution to make the results more convincing.
3)→In the Catalyst preparation part, for the sake of specification, please add a little more details about the preparation conditions, such as the purity of the drug and the source, as appropriate.
4)→In the XRPD of Co/SiO2 catalyst part, to show the change of catalyst, please add XRPD characterization of catalyst precursor, unreduced state and after reaction state.
5)→In the result part, to demonstrate the details of the acidic sites, please add the appropriate NH3-TPD, and CO2-TPD characterization , then further elaborate the actual situation of the acidic sites in the context of pyridine IR.
good
Author Response
Reviewer#2
In this manuscript, the authors carried on reaction evaluation of GVL hydrogenation utilizing pre-reduced Co/SiO2, discovered that partial reduction of the silica-supported cobalt oxide phase generated Co0 sites, active in hydrogenation/dehydrogenation reactions and the oxide residue represents Lewis acid (Co2+ ion)/Lewis base (O2- ion) pair sites, active in dehydration. As far as I'm concerned, there are still some points that could be further enhanced. So, I suggest a revision of the manuscript before publication. The recommendations are listed as below:
1)→The background introduction is a little overly brief, I hope that the background introduction can be more concise and detailed.
The introduction section was somewhat extended to show some more details of the background of the present study (see also our response given to Referee#3). Nevertheless, it should be mentioned that the relevant literature discussing the minor reaction routes in the hydroconversion of GVL to 2-MTHF is rather limited and/or they are mostly related to homogeneous phase batch reaction systems.
2)→In the Abstract part , please cite the detailed data of product distribution to make the results more convincing.
We understand Referee’s comment; however, since the product distribution changed in broad range, depending on the reaction parameters (especially in the function of space time) and on the reactant used (GVL, 2-MTHF, and 1,4-PD), we found it extremely difficult to show detailed product distributions in the Abstract. Instead, we summarized the most important conclusions concerning the minor pathways accompanied the GVL hydroconversion to 2-MTHF main product.
3)→In the Catalyst preparation part, for the sake of specification, please add a little more details about the preparation conditions, such as the purity of the drug and the source, as appropriate.
The source and purity of cobalt nitrate used for catalyst preparation is now given in the experimental section (4.1.). Some further details were also added to this section.
4)→In the XRPD of Co/SiO2 catalyst part, to show the change of catalyst, please add XRPD characterization of catalyst precursor, unreduced state and after reaction state.
The XRPD pattern of the unreduced catalyst precursor is already shown in Fig. 1a. We added also the pattern of the used catalyst (see in the supplemental material) upon Referee’s request. Note that the catalyst structure did not suffer any significant change during the reaction.
5)→In the result part, to demonstrate the details of the acidic sites, please add the appropriate NH3-TPD, and CO2-TPD characterization , then further elaborate the actual situation of the acidic sites in the context of pyridine IR.
The silica support used in the present study can be considered as inert (neither acidic nor basic) material, whereas introduction of Co resulted in the generation of Lewis acid sites (see section 2.4). Therefore, we did not considered CO2-TPD characterization. The NH3-TPD measurement is a useful technique for acidity characterization, if the origin of desorption peaks can be clearly identified (see, e.g. our former study in Micropor. Mesopor. Mater. 47 (2001) 293-301). However, the interpretation of the NH3-TPD spectrum becomes very difficult, when ammonia H-bonded to surface OH-groups also contribute to the TPD signal in the low temperature range. The presence of H-bonded base on the neat silica surface is obvious from the results shown in Fig. 4. Desorption of H-bonded species leads to the overestimation of real acid sites, which is a frequent problem in the relevant literature. In contrast, we found that the investigation of the FT-IR spectra of adsorbed pyridine was more suitable to distinguish pyridine interacting with surface silanols from those adsorbed on Lewis acid sites generated by the incompletely reduced fraction of Co-oxide on the silica surface. These Lewis acid sites are relatively strong as they could withhold Py up to 400 °C. Note that using the previously determined molar extinction coefficient (see Ref. [31]) the concentration of the real Lewis acid adsorption sites also could be calculated.
Reviewer 3 Report
The work presented by the authors is of great interest but it need of some modifications:
1) The authors should better comment the literature adding some examples of the employed catalysts. Which are the most adopted metal? Which acid catalysts have been employed? Which are the reaction conditions (temperature and hydrogen pressure)? How these parameters can generally influence the selectivity of the reaction?
2) The authors hypothesized the presence of 2-hydroxy-5-methytetrahydrofuran as an intermediate. Did they identify this intermediate?
3) 2-MTHF has not a lactone ring but a tetrahydrofuran ring
4) Control typo errors.
5) In my opinion, in order to better understand the observed trends, the results and discussion sections should be fused together.
6) The authors should test also different catalyst sizes in order to investigate the presence of mass transfer limitations and if the observed rate is the intrinsic rate of the reaction or it is influenced by the mass transfer rate.
7) The authors should also investigate the influence of hydrogen pressure.
Some errors are present in the manuscript, the authors should control the text.
Author Response
Reviewer#3
The work presented by the authors is of great interest but it need of some modifications:
1) The authors should better comment the literature adding some examples of the employed catalysts. Which are the most adopted metal? Which acid catalysts have been employed? Which are the reaction conditions (temperature and hydrogen pressure)? How these parameters can generally influence the selectivity of the reaction?
The introduction section was extended to give more details about the catalytic systems including reaction conditions formerly used in the title reaction. However, we note here that the relevant literature discussing the minor reaction routes in the hydroconversion of GVL to 2-MTHF is rather limited and/or they are mostly related to homogeneous phase batch reaction systems.
2) The authors hypothesized the presence of 2-hydroxy-5-methytetrahydrofuran as an intermediate. Did they identify this intermediate?
The 2-hydroxy-5-methytetrahydrofuran intermediate could not be identified (did not appear) in the product mixture; however, in a former study (see ref. [15]) we provided in situ DRIFT spectroscopic evidence for the hydrogenation of the carbonyl group indicating the formation of this surface intermediate. Other studies also assumed the formation of this intermediate (see e.g., refs. [3,12,18]).
3) 2-MTHF has not a lactone ring but a tetrahydrofuran ring
We are sorry for this mistake. It was corrected in the text (page 10).
4) Control typo errors.
The manuscript was checked thoroughly for typos.
5) In my opinion, in order to better understand the observed trends, the results and discussion sections should be fused together.
We agree, that sometimes it is easier to follow a combined “results and discussion” section. However, in this case we followed the instructions of the journal requiring separate “results” and “discussion” sections.
6) The authors should test also different catalyst sizes in order to investigate the presence of mass transfer limitations and if the observed rate is the intrinsic rate of the reaction or it is influenced by the mass transfer rate.
The absence of internal (intraparticle) diffusion limitation was verified by investigating the Weisz-Prater Criterion (CWP) for the particle size range used in the present study (0.315-0.63 mm). The results of the calculations, summarized in Table S2 in the supplemental materials, clearly show that diffusion limitations can be excluded in the catalytic system used in the present study.
7) The authors should also investigate the influence of hydrogen pressure.
The effect of hydrogen partial pressure was actually investigated; however, for the sake of simplicity, we originally decided to omit these results from the manuscript. Now, we added a new figure (together with a brief discussion) to the supplemental materials (see Fig. S3) showing the effect of hydrogen partial pressure on the reaction. The pressure dependence is also mentioned in section 2.5.1 in the manuscript.
Round 2
Reviewer 3 Report
The authors have modified the manuscript according to the reviewer's suggestion, thus it is ready for publication.